# Genetic Association between Farrowing Rates and Swine Leukocyte Antigen Alleles or Haplotypes in Microminipigs

**DOI:** 10.3390/cells11193138

**Published:** 2022-10-05

**Authors:** Asako Ando, Tatsuya Matsubara, Shingo Suzuki, Noriaki Imaeda, Masaki Takasu, Atsuko Shigenari, Asuka Miyamoto, Shino Ohshima, Yoshie Kametani, Takashi Shiina, Jerzy K. Kulski, Hitoshi Kitagawa

**Affiliations:** 1Department of Molecular Life Science, Division of Basic Medical Science and Molecular Medicine, Tokai University School of Medicine, Isehara 259-1193, Japan; 2Department of Veterinary Medicine, Faculty of Applied Biological Sciences, Gifu University, Gifu 501-1193, Japan; 3Laboratory of Veterinary Internal Medicine, Faculty of Veterinary Medicine, Okayama University of Science, 1-3 Ikoino-oka, Imabari 794-8555, Japan

**Keywords:** swine leukocyte antigen, reproductive performance, farrowing rate, haplotype, Microminipig

## Abstract

We have previously reported specific swine leukocyte antigen (*SLA*) haplotype associations with significant effects on several reproduction performance traits in a highly inbred miniature pig population of Microminipigs (MMPs). In this study, to clarify the effects on farrowing rates of SLA similarity between mating partners in the MMP population, we compared the farrowing rates as a measure of reproductive success after 1063-cumulative matings among the following three groups of mating partners: (1) completely sharing *SLA* class I or class II haplotypes or alleles between partners (CS), (2) only one sharing the haplotypes or alleles (OS), and (3) non-sharing the haplotypes or alleles (NS). Average farrowing rates in CS groups consisting of completely sharing *SLA* class II haplotypes or *DRBI* and *DQB1* alleles were lowest in the three groups. Moreover, lower farrowing rates were indicated in mating pairs with smaller amino acid pairwise genetic distances of *SLA-1*, *SLA-3*, *DRB1* and *DQB1* alleles between the pairs. These results suggested that the dissimilarity of *SLA* class I and class II alleles between mating partners markedly improved reproductive performance; therefore, *SLA* alleles or haplotypes are potentially useful genetic markers for the selection of mating pairs in breeding programs and epistatic studies of reproductive traits of MMPs.

## 1. Introduction

Many genetic and environmental factors play an important role for successful fertilization and conceptus growth through the gestation period, culminating in the birth of healthy offspring. Successful programs for breeding pigs from conception to birth depend on the outcomes of mating selected sows and sires, fertilization, implantation, gestation, and farrowing achievements. [1,2]. 

In pigs, as well as in other mammals such as mice, cattle, and humans, genes within the major histocompatibility complex (*MHC*) genomic region are one of the genetic factors involved in fertilization, selective abortions, and modulating mate choice [3,4,5]. Many studies showed that the sharing of certain maternal–fetal or paternal human leukocyte antigens (HLA) and bovine leukocyte antigens (BoLA) may influence fetal development and survival [5,6]. In pigs, the swine leukocyte antigen (*SLA*) class I and class II genes within the *SLA* genomic region are highly polymorphic and have important roles in the regulation of immune recognition and immune responses to foreign antigens and allo- or xeno-grafts in transplantation [7,8,9,10,11,12,13]. The polymorphisms of *SLA* genes enabled the analysis of associations between genotypes, alleles, haplotypes, and various immune responses and reproductive performance [7,14]. The influence of *SLA*-encoded genes on reproductive performances in domestic pigs has been reported as the association between specific *SLA* haplotypes and genital tract development in males, ovulation rates, litter sizes, and piglet weights at birth and weaning [7,15,16,17,18]. In selectively bred Duroc pigs with two low resolution *SLA* class II haplotypes (Lr), Lr-0.13 and Lr-0.30, assigned by a PCR-SSP method, Lr-0.30 was associated with higher weaning and rearing rates [16]. Furthermore, in a Landrace pig line with eleven *SLA* class II haplotypes, pigs with Lr-0.23 or Lr-0.13 had less severe pathological lesions of mycoplasmal pneumonia, increased leukocyte phagocytic activity, and higher white blood cell counts in comparison to pigs with other haplotypes [14]. 

A Microminipig (MMP) was developed by Fuji Micra Inc. (Fujinomiya, Shizuoka, Japan) as a novel miniature pig with an extremely small body size for use in laboratory biological and medical research. The body sizes, such as body weight, height, chest width, and chest circumference at 4–6 months of age, were much smaller than those of young mature beagles at 10 months old [17,18,19,20]. In the population of MMPs, 11 *SLA* class I and II haplotypes, including three recombinant haplotypes, were identified in the 14 parents or the progenitors of the highly inbred MMP herd. A total of 25 class I alleles (nine alleles of the *SLA-1*, eight alleles of the *SLA-2* and eight alleles of the *SLA-3* genes) were identified previously in MMPs with 8 *SLA* class I haplotypes. Homozygous MMPs with high resolution haplotype (Hp-) 35.0 had two different alleles at the *SLA-1* gene, *SLA-1^*^12:01* and *SLA-1^*^13:01,* suggesting duplicated *SLA-1* genes [21]. Recently, in the MMP population, we analyzed associations between the *SLA* class II haplotypes and reproductive performances, such as the fertility index that is expressed as the farrowing rate (reproductive success), gestation periods, litter sizes, and stillbirth rates in MMPs [18,22]. Two *SLA* class II haplotypes, Lr-0.13 and Lr-0.18, in both dams and sires showed lower and higher fertility indices, respectively, in comparison to the other six haplotypes. A significant effect of Lr-0.23 in dams also showed smaller litter size in comparison to seven other haplotypes [18]. Moreover, the *SLA* complex also appears to influence stillbirth rates in the MMP population [22].

In the present study, to evaluate the contribution of *SLA* class I and class II genes on reproductive traits in MMPs, we investigated the involvement of SLA sharing between mating pairs on their farrowing rates among *SLA* class I and class II haplotypes, including *SLA*-homozygous individuals. Furthermore, relationships among farrowing rates and amino acid distances of *SLA* alleles between mating pairs were also analyzed.

## 2. Materials and Methods

### 2.1. Animals

In this study, we used an MMP herd bred at Fuji Micra Inc. (Fujinomiya, Japan) from June 2008 to February 2017. In the herd, the records of 1063 cumulative matings of MMPs consisting of 114 sows and 44 boars assigned to 11 different *SLA* class I and class II haplotypes including 3 recombinant ones were utilized for the measurement of farrowing rates (reproductive success). The matings of MMPs were basically random. However, during some generations, especially with initial matings, mating pairs with relatively small body sizes were preferentially selected to establish the characteristics of the MMP breed [19]. 

This study was approved by the Animal Care and Use Committee of Gifu University (#17042, 26 May 2017). The care and use of the laboratory animals were in compliance with the guidelines of Good Laboratory Practice of Gifu University and Fuji Micra Inc.

### 2.2. SLA Class I and Class II Typing

Polymorphic *SLA* alleles for three class I (*SLA-1*, *SLA-2,* and *SLA-3*) and two class II (*DRB1* and *DQB1*) genes were assigned according to low-resolution *SLA* genotyping in 158 MMPs (114 sows and 44 boars), using a PCR-sequence-specific primers (SSP) method, as described previously [21]. Eleven types of high-resolution *SLA* class I and class II haplotypes (Hp-6.7, Hp-10.11, Hp-16.16, Hp-17.17, Hp-20.18, Hp-31.13, Hp-35.23 and Hp-43.37) including three class I and class II recombinant haplotypes (Hp-10.23, Hp-35.17 and Hp-43.17) were deduced from the preliminary results in the herd that were determined by performing an analysis of the inheritance and segregation of alleles of the three class I (*SLA-1*, *SLA-2*, and *SLA-3*), and two class II genes (*DRB1* and *DQB1*), respectively, in descendants of the MMP population (Table 1). To analyze the association between *SLA* class I, class II alleles, haplotypes and farrowing rates, and evolutionary amino acid divergences among *SLA* class I and class II alleles, the *SLA* alleles in four-digit genotypes of the 158 MMPs were deduced from the two-digit genotypes for the three class I (*SLA-1*, *SLA-2*, and *SLA-3*), and two class II (*DRB1* and *DQB1*) genes and the eleven types of low-resolution *SLA* class I and class II haplotypes (Table 1). 

### 2.3. Measurement of Farrowing Rates

To clarify the influence of the *SLA* allele sharing on farrowing rates, each mating pair was classified into one of three mating groups, CS, OS, and NS, that may or may not have shared *SLA* class I or class II alleles and haplotypes between sows and boars. Firstly, the completely shared (CS) alleles and haplotypes group consisted of mating pairs that completely shared *SLA* class I or class II alleles and haplotypes encoding the *SLA-1*, *SLA-2*, *SLA-3*, *DRB1* and *DQB1* genes. Secondly, the only-one-shared (OS) group, consisted of mating pairs that shared only one *SLA* class I or class II allele or haplotype. Thirdly, the non-shared (NS) *SLA* alleles and haplotypes group consisted of mating pairs that had a completely different *SLA* class I or class II alleles and haplotypes. 

Average farrowing rates in the three mating groups with different shared *SLA* haplotypes or alleles between sows and boars were calculated as the ratios of the number of deliveries including both live and still births compared to the number of matings. The farrowing rate in each mating group was calculated as the ratio of the number of deliveries to that of matings. The farrowing rates were compared among the three mating groups with the different shared *SLA* alleles or haplotypes.

### 2.4. Influence of Amino Acid Distance of SLA Class I and Class II Genotypes between Mating Partners on Farrowing Rates 

To evaluate the relationship between farrowing rates and amino acid divergences of *SLA* class I (*SLA-1*, *SLA-2,* and *SLA-3*) or class II (*DRB1* and *DQB1*) alleles in each mating pair, amino acid pairwise distances among the alleles were analyzed with MEGA X software using the JTT matrix-based model with a gamma distribution (Appendix A). The pairwise distances were calculated based on the number of amino acid substitutions per site between their allele sequences [23,24]. The sum of all the pairwise amino-acid distances (D) of the four possible alleles (A, a, B, and b) in each mating pair was calculated as follows:

Alleles with sire: A, a;

Alleles with dam: B, b.

The sum of all the pairwise amino-acid distances = (D_AB_ + D_Ab_ + D_aB_ + D_ab_). If the mating pair has exactly the same alleles, the sum of all the pairwise amino-acid distances is ‘0’.

Farrowing rates were calculated as the ratios of the number of deliveries to those of matings in range groups that were divided according to the sum of all the amino acid pairwise distances in each *SLA* locus among the possible four alleles between mating partners shown in Table 2. The sums of the amino acid pairwise distances among four alleles in each of the *SLA* class I genes and class II-*DRB1* genes carried by each mating pair were classified into nine to eleven groups, which increased in distance from each other in intervals of 0.1. On the other hand, the sum of the amino acid pairwise distances among the *SLA* four alleles of the class II- *DQB1* gene carried by each mating pair was classified into eight groups that increased in distance from each other by 0.05 ranges. Correlation coefficients were calculated between the central value of each range group and farrowing rate.

### 2.5. Statistical Analyses 

Farrowing rates indicated as absolute numbers were evaluated by the Chi-square for independence test, using an m×n contingency table (BellCurve in Excel, Social Survey Research Information Co., Ltd. Tokyo, Japan). In Figure 1, Figure 2, Figure 3 and Figure 4, the farrowing rates were indicated as percentages for simpler comparisons. Correlations among farrowing rates and amino acid pairwise distances of *SLA-1*, *SLA-2*, *SLA-3*, *DRB1* and *DQB1* alleles between mating partners were evaluated by Spearman’s rank correlation coefficient. *p*-values of less than 0.05 were considered significant.

## 3. Results

### 3.1. Association between Farrowing Rates and Sharing of SLA Class I Haplotypes or Alleles

A total of 656 pregnancies were obtained as the result of 1063 matings of 158 MMPs consisting of 114 sows and 44 boars, representing a farrowing rate of 61.7%. The farrowing rates in MMPs was considerably lower than 88.4 ± 4.6 (standard deviation (SD)) in mixed breed domestic pigs in Japan [25]. 

In the present study, 25 *SLA* class I alleles including two *SLA-1* alleles (*SLA-1^*^12:01* and *SLA-1^*^13:01*) encoded by duplicated *SLA-1* genes in 114 dams and 44 sires were inherited as eight class I haplotypes without cross over among the three class I loci, *SLA-1*, *SLA-2*, and *SLA-3*. Therefore, the eight *SLA* class I haplotypes in MMPs consist of nine, eight, and eight classical *SLA* class I alleles in the *SLA-1*, *SLA-2,* and *SLA-3* loci, respectively. Since associations among farrowing rates and the sharing of *SLA* class I haplotypes amount to the same results as association among farrowing rates and the sharing of *SLA* class I alleles, the following farrowing rates were found mostly as the sharing of *SLA* class I haplotypes.

The average farrowing rates of 114 dams of MMPs in the three mating groups, CS, OS, and NS, as classified according to the sharing of *SLA* class I haplotypes between partners, were 58.5%, 61.2% and 63.8%, respectively. 

The farrowing rate of the CS group was lowest in comparison to those of the other two groups, the OS, and NS. However, no significant differences of the farrowing rates were obtained among the three groups consisting of different shared *SLA* class I haplotypes (Figure 1A). Furthermore, in the comparison of the farrowing rates between the CS group and other mating groups, no significant differences of the farrowing rates were obtained between the two groups, the CS and (OS + NS), consisting of only one sharing or non-sharing *SLA* class I haplotype between partners (Figure 1B).

### 3.2. Association between Farrowing Rates and Sharing of SLA Class II Haplotypes or Alleles

In addition to the three *SLA* class I recombinant haplotypes, there were only eight distinct *SLA* class II haplotypes in this MMP population (Table 1). These were Hp-0.7, Hp-0.11, Hp-0.13, Hp-0.16, Hp-0.17, Hp-0.18, Hp-0.23 and Hp-0.37, as determined by an analysis of the inheritance and segregation of eight and four genotypes of the *DRB1* and *DQB1* genes, respectively. Thus, we used the *DRB1* and *DQB1* alleles individually as well as their haplotypic linkages to assess their effects on farrowing rates. Since each of the *DRB1* alleles corresponds specifically to only one of the eight class II haplotypes in the MMP population (Table 1), we used the *DRB1* alleles as genetic markers of the class II haplotypes.

The 1063 matings of 158 MMPs were classified into the following three groups: the CS, OS, and NS groups, according to the sharing of *SLA* class II-*DRB1* or -*DQB1* alleles between each mating pair. The average farrowing rates in the three mating groups, CS OS, and NS, classified according to the sharing of *DRB1* alleles between dams and sires in each mating pair, were 53.0%, 63.5% and 61.7%, respectively. The average farrowing rate of the CS group was lowest when compared to those of the other two groups, the OS and NS groups. The average farrowing rate of the CS group was significantly lower than those of OS and NS groups (*p* = 0.0165 and *p* = 0.0484, respectively (Figure 2A)). Furthermore, the average farrowing rate of the CS group was also significantly lower than that of the (OS + NS) group consisting of only one sharing and non-sharing *DRB1* with completely different alleles between partners (*p* = 0.0198 (Figure 2B)).

The average farrowing rates in the three mating groups that were classified according to the sharing of *DQB1* alleles between dams and sires in each mating pair were 54.1%, 63.7% and 61.6%, respectively. The average farrowing rate of the CS group was lowest in comparison to those of the other two groups, the OS and NS groups. A significant difference was observed between average farrowing rates of CS group and OS group (*p* = 0.0139 (Figure 3A)). When the average farrowing rates were compared between the CS and (OS + NS) groups, a significant difference was also observed (*p* = 0.0172 (Figure 3B)).

In comparing the sharing of *SLA* class II haplotypes and *DRB1* alleles between dams and sires, the mating pairs were classified also into the CS, OS, and NS groups. The average farrowing rates of dams in CS groups with completely shared *DRB1* or *DQB1* genotypes between dams and sires were lowest when compared with the other mating groups with different class II alleles (Figure 2A and Figure 3A).

### 3.3. Effects of Amino Acid Pairwise Distances between SLA Class I Alleles of Mating Pairs on Farrowing Rates in MMPs

Amino acid distances among nine *SLA-1*, eight *SLA-3* and eight *SLA-2* alleles of mating pairs showed various values depending on the class I loci (Appendix A). The overall mean distances among the *SLA-1*, *SLA-3* and *SLA-2* alleles were 0.1972, 0.1178, and 0.2330, respectively. Since MMPs with Hp-35.0 duplicated *SLA-1* genes that encode two alleles named as *SLA-1^*^12:01* and *SLA-1^*^13:01*, the sum of pairwise amino acid distances of *SLA-1* alleles of mating pairs was calculated separately for *SLA-1^*^12:01* and *SLA-1^*^13:01* (Table 2). The sum of the pairwise distances in 1063 matings among four *SLA-1* alleles between partners increased from 0 to 0.910, and 0 to 0.994 using the pairwise distances with *SLA-1^*^12:01* and *SLA-1^*^13:01*, respectively, when either or both of the partners had Hp-35.0. These pairwise distances among four *SLA-1* alleles between partners were classified into ten and eleven groups that were divided across 0.1 ranges using the pairwise distances with *SLA-1^*^12:01* and *SLA-1^*^13:01*, respectively. As shown in Figure 4A,B, lower farrowing rates were indicated in mating pairs with relatively smaller pairwise genetic distances of *SLA-1* alleles. A significant correlation was observed by Spearman’s correlation coefficient analysis (*p* = 0.0320) in Figure 4B among farrowing rates and amino acid pairwise distances of *SLA-1^*^13:01* between the mating pairs.

The sum of the pairwise distances in 1063 matings among *SLA-3* alleles between partners increased from 0 to 0.718, and was classified into nine groups that were divided across 0.1 intervals (Table 2). As shown in Figure 4C, lower farrowing rates were indicated in mating pairs with smaller pairwise genetic distances of *SLA-3* alleles. A significant correlation among farrowing rates and amino acid distances of *SLA-3* alleles between partners was observed (*p* = 0.0501 (Figure 4C)). The sum of the pairwise distances among *SLA-2* alleles between partners increased from 0 to 1.154, and was classified into eleven groups that were divided across 0.1 ranges (Table 2). There was no significant differences among the farrowing rates and amino acid distances of *SLA-2* alleles between the mating pairs (Figure 4D).

### 3.4. Effects of Amino Acid Pairwise Distances between SLA Class II Alleles of Mating Pairs on Farrowing Rates in MMPs

As shown in Appendix A, amino acid distances among eight *DRB1* alleles of mating pairs showed relatively high values, and the overall mean distance among the *DRB1* alleles was 0.1695. The sum of the amino acid pairwise distances in each of the 1063 matings among *DRB1* alleles was classified into ten groups from 0 to 0.899 across 0.1 ranges (Table 2). The ranges of the sum of the pairwise distances from 0 to 0.831 among *DRB1* alleles between partners were comparable with those of the class I *SLA-1*, *SLA-2*, and *SLA-3* alleles between partners (Table 2). In contrast, the differences of the amino acid pairwise distances among six kinds of *SLA* class II-*DQB1* alleles between mating partners ranging from 0.0235 to 0.0867 (overall mean distance: 0.0651) were at a relatively low level when compared with those differences of the *SLA-1*, *SLA-2*, *SLA-3*, and *DRB1* alleles with the mating partners (Appendix A). Due to the narrow distribution of pairwise distances and small number of different *DQB1* alleles, the sum of the amino acid pairwise distances in each of the 1063 matings among *DQB1* alleles was classified into only eight groups that ranged from 0 to 0.390 with ranges of 0.05 (Table 2). As shown in Figure 3A,B, lower farrowing rates were indicated in mating pairs with smaller pairwise genetic distances of *DRB1* or *DQB1* alleles. Significant correlations among farrowing rates and amino acid distances of *DRB1* and *DQB1* alleles between partners were observed by Spearman’s correlation coefficient analysis (*p* = 0.0118 (Figure 4E) and *p* = 0.0102 (Figure 4F)) for *DRB1* and *DQB1* alleles, respectively. 

## 4. Discussion

The farrowing rate in MMPs of 61.7% was considerably lower than in mixed breed domestic pigs in Japan [25]. The reason why the farrowing rate in MMPs was lower than in the mixed breed domestic pigs is not known. However, other reproductive performances such as gestation periods, litter sizes at birth, and weaning in MMPs [18] were similar to those of other breeds of domestic pigs [15,25,26] including Göttingen and NIBS minipigs [27,28]. Thus, apart from the farrowing rates, the MMP population used in the present study appears to have relatively normal porcine reproductive traits.

In this study, eleven *SLA* class I and class II haplotypes including three recombinant haplotypes, Hp-10.23, Hp-35.17 and Hp-43.17, were found in 228 dams and 88 sires of MMPs (Table 1), which is consistent with our previous *MHC* haplotyping results of the MMP population. In the three recombinant haplotypes, all of the DNA crossover regions in the haplotypes were found within the class III region [21]. Frequencies of dams and sires for the mating performance with three recombinant haplotypes, Lr-10.23, Lr-35.17, and Lr-43.17, were found to be between 0% and 8% (Table 1). Due to the mating pairs with the recombinant haplotypes, the numbers of three mating groups on *SLA* class I haplotypes, CS, OS, and NS, were different from those on *SLA* class II haplotypes. Therefore, we separately analyzed the average farrowing rates of mating partners sharing *SLA* class I and class II haplotypes. The average farrowing rates of mating partners sharing *SLA* class II haplotypes *DRB1* and/or *DQB1* alleles were relatively low compared to those of non-sharing haplotypes. Since the two *p* values (*p* = 0.0198 and *p* = 0.0172) that were obtained from the association analyses between farrowing rates and the sharing of *DRB1* and *DQB1* showed almost the same level, it is difficult to determine which gene, *DRB1* or *DQB1*, has higher significant effects for better farrowing rates. In contrast to the *DRB1* and the *DQB1* results, the degree of sharing *SLA* class I haplotypes or alleles between mating partners had no effect on the average farrowing rates. Taken together, these results suggested that the dissimilarities of *SLA* class II haplotypes or alleles between partners could provide more benefits on farrowing rates than that of *SLA* class I haplotypes or alleles. Nevertheless, we analyzed the genetic association between farrowing rates and *SLA* alleles or haplotypes using a relatively small number of MMPs in this study. Therefore, it might be difficult to exclude the possibilities of biased results in the comparison of farrowing rates between sharing or non-sharing *SLA* alleles or haplotypes. The reliability of the present results could be confirmed in future studies by using a larger number of MMPs and other pig breeds.

Moreover, higher farrowing rates were observed in mating pairs with bigger amino acid pairwise genetic distances of *SLA-1* alleles using those of *SLA-1^*^13:01*, *SLA-3*, *DRB1* or *DQB1* alleles between the pairs. Relatively higher farrowing rates were also observed in mating pairs with bigger amino acid pairwise genetic distances of *SLA-1* alleles using those of *SLA-1^*^12:01* or *SLA-2* alleles between the pairs, although no obvious significant correlations among farrowing rates and amino acid distances of *SLA-1* alleles using those of *SLA-1^*^12:01* or *SLA-2* alleles between partners were observed by Spearman’s correlation coefficient analysis. A high farrowing rate, 90%, was obtained for the sum of the amino acid pairwise distances of *SLA-2* between 0.3 and 0.399, although this may be due to a potential bias using only 20 matings which suggests that the number of matings might have an influence on the correlation coefficient analysis (Table 2, Figure 4D). Nevertheless, these results revealed that amino acid dissimilarities of *SLA-1*, *SLA-3*, *DRB1* and *DQB1* alleles between mating pairs had significant effects on the farrowing rates in MMPs.

Relationships between farrowing rates and *MHC*-dissimilarity or *MHC* heterozygosity have been reported in mate choice studies on many kinds of animals including rodents [3,4,29], humans [30,31], non-human primates [31], horses [32], birds [33] and fish [34]. Namely, in the theory of mate choice based on heterozygosity, partners with MHC-dissimilarity have a greater tendency to choose each other than those with MHC-similarity. The preference for MHC-dissimilarity between potential partners may function to avoid inbreeding and promote offspring heterozygosity for greater resistance to pathogens [35,36]. Our present results also showed that the dissimilarities of *SLA* class I and class II alleles between the pairs might be involved in farrowing rates in the MMP population. Thus, our findings in this and previous studies of reproduction in the MMP population [16] support the overall hypothesis that *MHC* heterozygosity enhances reproductive success. However, we have not analyzed whether the offspring heterozygosity would be promoted by MHC-dissimilarity between potential partners in the MMPs. Recently, a tendency for humans to prefer MHC-dissimilar mates was also demonstrated by using *HLA* types and 9,010 single nucleotide polymorphisms (SNPs) densely distributed across the MHC in 30 European American couples [37]. Furthermore, the same research group reported that the MHC influences mate choice, although social constraints in some populations also affect the mate choice [38]. On the other hand, meta-analyses in humans focusing on genomic mate selection, relationship satisfaction, odor preference and relationship satisfaction revealed that there was no association between MHC-dissimilarity and mate choice in actual couples [39]. Nevertheless, to clarify the mechanism of MHC-based mating preference, further genome wide association studies (GWAS) will be necessary to identify candidate genes correlated with mate choice and the effects of epistasis.

In the MMP population, *SLA* haplotype sharing and amino acid dissimilarities of *SLA* class I or class II alleles between partners had a weak influence on their farrowing rates. In general, the overall effects of MHC-based mating preferences are relatively weak. For instance, in the overall effect sizes of primates, Fisher’s Z correlation coefficient for dissimilarity showed Zr = 0.044 [31]. Furthermore, small effect sizes on mate choice for MHC-dissimilarity in non-human vertebrates were also found using formal phylogenetic meta-analysis and meta-regression techniques. 

Additionally, the effect of MHC dissimilarity in mating preference was found when the dissimilarity was characterized at multiple loci including *MHC* class I and II loci [40]. Therefore, the effect of MHC dissimilarity by multiple loci is consistent with our results that dissimilarities of *SLA-1*, *SLA-3*, *DRB1* and *DQB1* alleles between mating pairs had significant effects on the farrowing rates in MMPs. 

In humans, non-classical MHC class I molecules, HLA-E and HLA-G, can interact to maintain immune homeostasis of the maternal–fetus interface with many kinds of receptors on maternal immune cells in the uterus. HLA-E is the ligand of inhibitory and activating receptors, CD94/NKG2A and CD94/NKG2C, respectively, and HLA-G is the ligand of inhibitory receptors, ILT-2 (CD85j/LILRB1) and ILT-4 (CD85d/LILRB2) expressed on natural killer (NK) cells [41,42]. The pig appears to have many novel LILR genes, and only one KIR gene [43], but the specific gene products that are expressed and which play a role in porcine reproduction is still not known. Regarding the achievement of porcine deliveries, there are many biological and genetic factors other than MHC-based mating preferences involved in the important processes from fertilization to successful deliveries. To facilitate porcine pregnancy success, many immune cells such as uterine NK cells, dendritic cells, and macrophages are involved in the regulation of placental development, homeostasis, and tolerance of the fetal allograft [2]. Uterine endometrial expression of SLA class I molecules in stromal cells and luminal epithelium cells has been analyzed at each pregnancy stage, and it is generally accepted that the expression of SLA class I and β-2 microglobulin (β_2_M) molecules decrease in the placenta to prevent conceptus as a semi-allograft from host-vs-graft immune rejection. Nevertheless, classical SLA class I (SLA-1, SLA-2, and SLA-3) and nonclassical SLA class I (SLA-6, SLA-7, and SLA-8) and β_2_M molecules increased in uterine stromal cells and luminal epithelium cells during the peri-attachment period; then, their abundant expression decreased after successful conceptus attachment. 

Cell-type specific regulation of the *SLA* and *B2M* genes is controlled by progesterone and IFNs on uterine cells [44]. However, the correlation between differential fertilization success depending on SLA sharing and the pregnancy stage specific expression of SLA class I molecules remains unclear. Regarding *SLA* class II expression during pregnancy, DQ molecules increased in response to conceptus-derived interferon gamma (IFNG) and likely regulate immune response at the maternal–fetal interface to support the maintenance of pregnancy in pigs [45]. Moreover, HLA class I and class II molecules were expressed in ejaculated spermatozoa, thereby suggesting some undefined roles of the HLA molecules on spermatozoa function and immune activation in female reproductive tracts. The expression of the HLA class II molecules was significantly higher than those of HLA class I in human spermatozoa [46]. In mice, it was suggested that the class II molecules on the posterior region of the sperm head plays an adhesive role in the recognition between sperm and egg during fertilization [47]. These reports on the expression of SLA class I and class II molecules suggested that the SLA molecules might be involved in porcine pregnancy success with many other immune factors. Here, we described the relatively weak and variable effects of the dissimilarity of *SLA* class I and II alleles between mating pairs on farrowing rates. Therefore, the *SLA* class I and class II genes or haplotypes might not be directly responsible for porcine pregnancy success with high farrowing rates. Although the polymorphic features of *SLA* genes or haplotypes and SLA dissimilarity between mating pairs might correlate only indirectly with the pregnancy process of MMPs; it nevertheless is evident from the present study that *SLA* class I and class II alleles or haplotypes could be useful genetic markers for the selection of mating pairs in breeding programs for assisting with a more effective breeding management of MMPs. 

Since the effects of SLA dissimilarity between mating partners on farrowing rates in our present study were analyzed using only a few *SLA* haplotypic class I and II loci in a limited pig population and breed, further studies are needed to confirm the effects of SLA dissimilarity on pregnancy and the achievement of deliveries using other pig breeds with other *SLA* haplotypes. 

## 5. Conclusions

Farrowing rates of mating pairs with shared *SLA* class II haplotypes or alleles were relatively lower than those with non-sharing ones. Furthermore, analyses between amino acid distances of *SLA* alleles between mating pairs exhibited that the dissimilarities of *SLA* class I and class II alleles might be involved in farrowing rates in the MMP population. These relationships between the farrowing rates and SLA suggested that the *SLA* class I and class II alleles or haplotypes have the potential to be useful genetic markers for the selection of mating pairs in breeding programs and epistatic studies of reproductive traits of MMPs.

## Figures and Tables

**Figure 1 cells-11-03138-f001:**
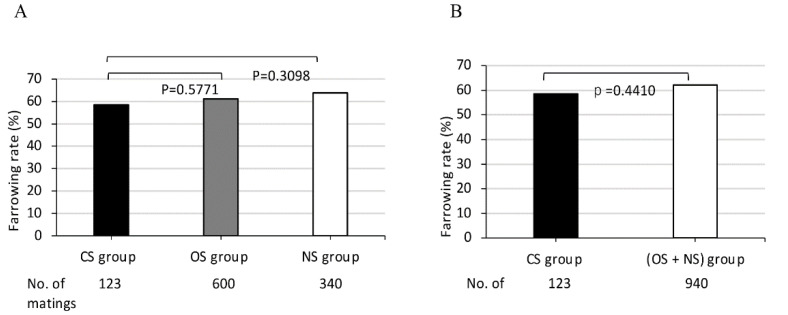
Comparison of farrowing rates in MMPs among (**A**) three groups CS, OS, NS, and (**B**) between two groups (**B**), CS and (OS + NS). CS, completely sharing *SLA* class I haplotypes between partners; OS, sharing only one *SLA* class I haplotype between partners; NS, non-sharing of *SLA* class I haplotypes between partners. X-axis shows CS, OS, and NS groups (**A**) and CS and (OS + NS) groups (**B**) and the number of matings for each group. Y-axis shows farrowing rate as indicated by the ratio (%) of the number of deliveries to the number of matings, expressed as the mean value (bar). Black and white bars represent lower and higher farrowing rate, respectively, of the mean values. Gray bars represent intermediate farrowing rate of the mean values between those of black and white bars.

**Figure 2 cells-11-03138-f002:**
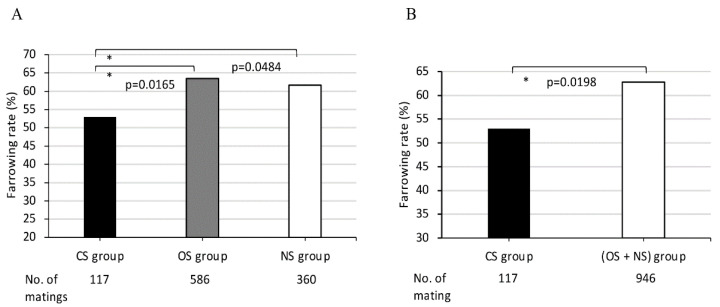
Comparison of farrowing rates in MMPs among (**A**) three groups CS, OS, NS, and (**B**) between two groups, CS and (OS + NS). CS, completely sharing *SLA* class II haplotypes or *DRB1* alleles between partners; OS, sharing only one *SLA* class II haplotype or *DRB1* alleles between partners; NS, non-sharing, that is, *SLA* class II haplotypes or *DRB1* alleles were completely different between partners. X-axis shows CS, OS, and NS groups (**A**), and CS and (OS + NS) groups (**B**), and the number of matings for each group. Y-axis shows farrowing rate as indicated by the ratio (%) of the number of deliveries to the number of matings, expressed as the mean value (bar). Black and white bars represent lower and higher farrowing rate, respectively, of the mean values. Gray bars represent intermediate farrowing rate of the mean values between those of black and white bars. Probabilities of significant differences among groups are indicated by single (*p* < 0.05) asterisks.

**Figure 3 cells-11-03138-f003:**
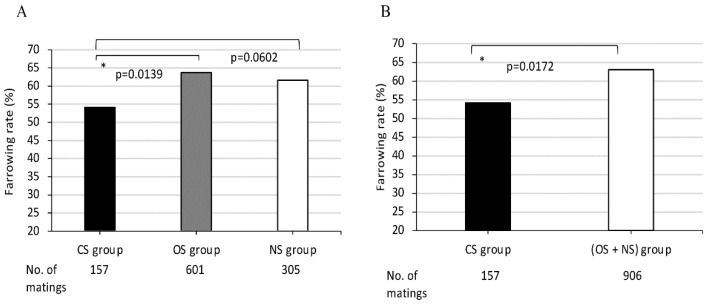
Comparison of farrowing rates in MMPs among (**A**) three groups CS, OS, NS, and (**B**) between two groups CS and (OS + NS). CS, completely sharing *SLA* class II-*DQB1* alleles between partners; OS, sharing only one *DQB1* allele between partners; NS, non-sharing, that is, completely different *DQB1* alleles between partners. X-axis shows CS, OS, and NS groups (**A**) and CS and (OS + NS) groups (**B**) and the number of matings for each group. Y-axis shows farrowing rate as indicated by the ratio (%) of the number of deliveries to the number of matings, expressed as the mean value (bar). Black and white bars represent lower and higher farrowing rate, respectively, of the mean values. Gray bars represent intermediate farrowing rate of the mean values between those of black and white bars. Probabilities of significant differences among groups are indicated by single (*p* < 0.05) asterisks.

**Figure 4 cells-11-03138-f004:**
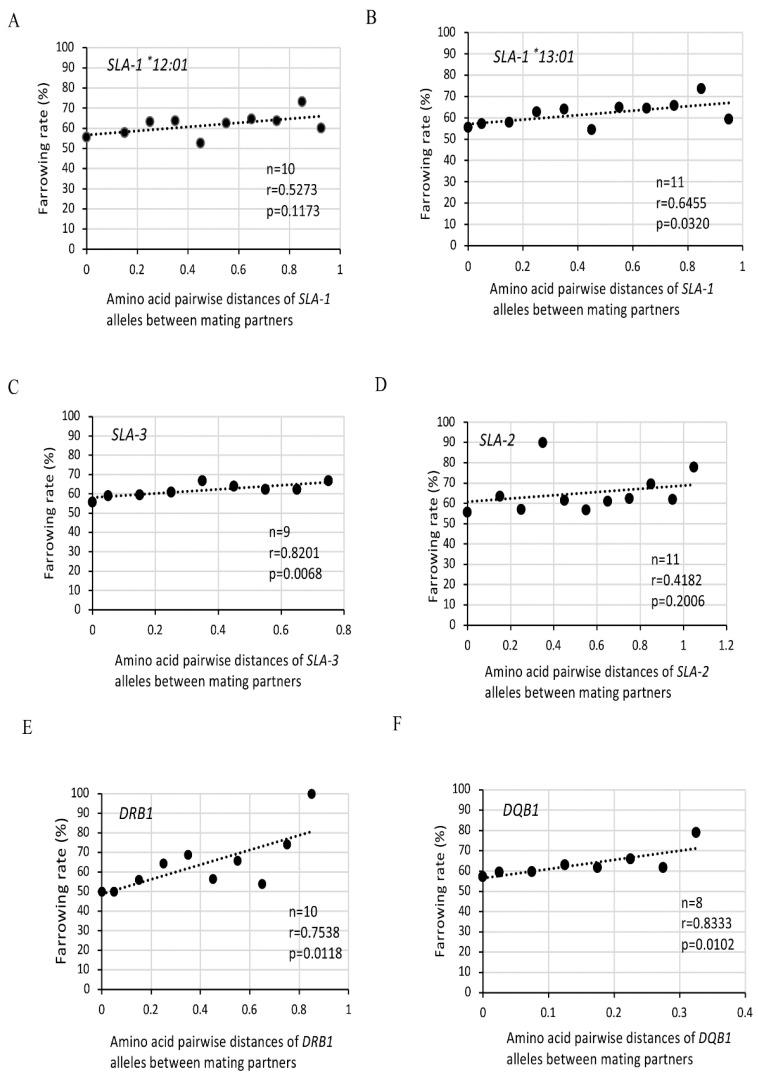
Effects of amino acid pairwise distances of *SLA* class I and class II alleles between partners on farrowing rates in MMPs. Dots are indicated as farrowing rates (%) in the center value of each group of amino acid pairwise distances of *SLA-1* (**A**,**B**), *SLA-3* (**C**), *SLA-2* (**D**), *DRB1* (**E**), and *DQB1* (**F**) alleles between partners as shown in Table 2. Dotted line shows the approximated line. n: number of groups on the sum of pairwise amino acid differences of *SLA-1* (**A**,**B**), *SLA-3* (**C**), *SLA-2* (**D**), *DRB1* (**E**), and *DQB1* (**F**) alleles between partners as shown in Table 2. r: correlation coefficient, and *p*: *p* value evaluated by Spearman’s rank correlation coefficient.

**Table 1 cells-11-03138-t001:** *SLA*-class I and II genotypes and number of *SLA* haplotypes in Microminipigs.

No.	Haplo-Type (Hp-)	*SLA*-Class I			*SLA*-Class II		Number of Haplotypes ^c)^			
							(Frequency (%))			
		SLA-1 ^a)^	SLA-3 ^a)^	SLA-2 ^a)^	DRB1 ^a)^	DQB1 ^a)^	Dams		Sires	
1	6.7	^*^08:05	^*^06:01	^*^05:04	^*^06:01	^*^06:01	1	(0.4)	0	(0)
2	10.11	^*^05:01	^*^08:01	^*^03:02	^*^09:01	^*^04:02	17	(7.5)	6	(6.8)
3	16.16	^*^04:01	^*^06:02	^*^09:01	^*^11:03	^*^06:01	14	(6.1)	6	(6.8)
4	17.17	^*^08:04	^*^03:05	^*^06:03	^*^08:01	^*^05:01	28	(12.3)	8	(9.1)
5	20.18	^*^10:02	^*^01:01	^*^11:01:02	^*^14:01	^*^04:01:02	19	(8.3)	4	(4.5)
6	31.13	^*^15:02	^*^07:01:02	^*^16:01	^*^04:03	^*^03:03	12	(5.3)	2	(2.3)
7	35.23	^*^12:01,^*^13:01	^*^05:02	^*^10:01	^*^10:01	^*^06:01	69	(30.3)	33	(37.5)
8	43.37	^*^11:04	^*^04:01	^*^04:02:02	^*^07:01	^*^05:02	56	(24.6)	20	(22.7)
9	10.23 ^b)^	^*^05:01	^*^08:01	^*^03:02	^*^10:01	^*^06:01	0	(0)	1	(1.1)
10	35.17 ^b)^	^*^12:01,^*^13:01	^*^05:02	^*^10:01	^*^08:01	^*^05:01	7	(3.1)	7	(8.0)
11	43.17 ^b)^	^*^11:04	^*^04:01	^*^04:02:02	^*^08:01	^*^05:01	5	(2.2)	1	(1.1)

^a)^ Allele specificity that is assigned by low resolution typing at two-digit level indicates as expected allele specificity by high resolution typing in Microminipigs. *DQB1^*^04:02* (Hp-0.11) and *DQB1^*^04:01:02* (Hp-0.18), and *DQB1^*^05:01* (Hp-0.17) and *DQB1^*^05:02* (Hp-0.37) are assigned as *DQB1^*^04:XX* (Lr-0.11 or Lr-0.18) and *DQB1^*^05:XX* (Lr-0.17 or Lr-0.37) using a PCR-SSP method, respectively. ^b)^ Hp-10.23, Hp-35.17, and Hp-43.17 were assigned as recombinant haplotypes [21]. ^c)^ The numbers of dams and sires are 228 and 88, respectively.

**Table 2 cells-11-03138-t002:** Number of matings in each range classified by the sum of the amino acid pairwise distances of SLA class I and class II alleles between mating pairs for correlation analyses among farrowing rates and the amino acid pairwise distances.

Amino Acid Pairwise Distances ^a)^	No. of Mating					Amino Acid Pairwise Distances ^b)^	No. of Mating
Range	SLA-1	SLA-1	SLA-3	SLA-2	DRB1	Range	DQB1
	^*^12:01 ^c)^	^*^13:01 ^c)^					
0	18	18	18	18	12	0	49
0.010–0.099	0	42	217	0	2	0.010–0.049	180
0.100–0.199	229	190	116	71	222	0.050–0.099	173
0.200–0.299	57	94	284	214	156	0.100–0.149	221
0.300–0.399	96	50	189	20	164	0.150–0.199	180
0.400–0.499	101	160	72	83	124	0.200–0.249	182
0.500–0.599	184	145	119	60	226	0.250–0.299	59
0.600–0.699	183	104	45	231	123	0.300–0.349	19
0.700–0.799	102	170	3	212	28		
0.800–0.899	78	53		82	6		
0.900–0.999	15	37		63			
1.000–1.199				9			

^a)^ The sum of the amino acid pairwise distances in the duplicated *SLA-1*, *SLA-3*, *SLA-2* or *DRB1* genes among the four possible *SLA* alleles between mating partners were divided into ten, eleven, nine, eleven or ten groups of ranges with a 0.1 interval, respectively. ^b)^ Due to the very narrow ranges of the amino acid pairwise distances among *DQB1* alleles, the sum of the amino acid pairwise distances among the possible four *DQB1* alleles between mating partners were divided into eight groups across 0.05 ranges. ^c)^ Due to the duplicated *SLA-1* genes encoding two alleles named as *SLA-1^*^12:01* and *SLA-1^*^13:01* in MMPs with Hp-35.0, the sum of the pairwise amino acid distances of *SLA-1* alleles was calculated separately in *SLA-1*12:01* and *SLA-1*13:01*.

## Data Availability

Not applicable.

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
