# Peer review of "Genetic Association between Farrowing Rates and Swine Leukocyte Antigen Alleles or Haplotypes in Microminipigs"

_cells, 2022, doi:10.3390/cells11193138_

Round 1

Reviewer 1 Report

The manuscript submitted by Ando et al. is a logical follow-up on their previous studies of the Microminipig model. Based on data on associations between MHC alleles/haplotypes and reproductive performance, including the farrowing rate, the authors have studied effects of allele/haplotype sharing on the farrowing rate. Besides effects of MHC sharing, they also have evaluated effects of amino acid distances of SLA alleles between mating pairs.

As a whole, the manuscript is well written and documented. However, some clarifications and amendments could improve it.

1.      Materials and Methods, section 5.2. It is not clear from the text whether percentages were used only for indicating farrowing rates, as written on line 152, and absolute numbers were used for calculating the Chi-square values, or whether the percentages were also used for Chi-square calculations. Please clarify in the text. If the percentages were used for calculations, please provide reasons for this approach.

2.      Results, section 3.1, lines 162-163. Possible reasons for the difference between MMPs and mixed breed domestic pigs could be briefly mentioned in Discussion.

3.      Results, sections 3.3. and 3.4. The text of the two sections is rather difficult to read. To improve the readability, their introductory parts could be moved to Materials and Methods, section 2.4.; it is up to the authors to judge whether data in the tables 2 to 7 could be substituted with a text summary of the findings and whether the tables could be attached as supplementary materials. In my view, at least the addition of a clearly formulated summary of the data contained in the tables would significantly contribute to the readability of this part of the manuscript.

4   The Discussion as a whole is evidence-based and well written. I recommend adding information on possible interactions of the maternal NK cell receptor KIR and fetal HLA class I molecules in human reproduction in that part of Discussion dedicated to the biological context of the data presented (perhaps between lines 456-460).

Author Response

For Reviewer 1

Comments and Suggestions for Authors

The manuscript submitted by Ando et al. is a logical follow-up on their previous studies of the Microminipig model. Based on data on associations between MHC alleles/haplotypes and reproductive performance, including the farrowing rate, the authors have studied effects of allele/haplotype sharing on the farrowing rate. Besides effects of MHC sharing, they also have evaluated effects of amino acid distances of SLA alleles between mating pairs.

As a whole, the manuscript is well written and documented. However, some clarifications and amendments could improve it.

  1. Materials and Methods, section 5.2.It is not clear from the text whether percentages were used only for indicating farrowing rates, as written on line 152, and absolute numbers were used for calculating the Chi-square values, or whether the percentages were also used for Chi-square calculations. Please clarify in the text. If the percentages were used for calculations, please provide reasons for this approach.

Response: Yes, we have used absolute numbers for calculation of the Chi-square values. Further, we have used the percentages in farrowing rates in the figures to simplify comparisons. Thus, we have revised sentences in Materials and Methods as follows,

  1. Materials and Methods

2.5. Statistical analyses (lines 170-173)

Farrowing rates indicated as absolute numbers were evaluated by the Chi-square for independence test, using an m×n contingency table (BellCurve in Excel, Social Survey Research Information Co., Ltd. Tokyo, Japan). In Figure 1-4, the farrowing rates were indicated as percentages for simpler comparisons.

  1. Results, section 3.1, lines 162-163. Possible reasons for the difference between MMPs and mixed breed domestic pigs could be briefly mentioned in Discussion.

Response: According to the Reviewer’s suggestion, we added following sentences in Discussion.

  1. Discussion (lines 447-453)

The farrowing rates in MMPs representing 61.7% was considerably lower than in mixed breed domestic pigs in Japan [25]. The reason why the farrowing rates in MMPs was lower than in the mixed breed domestic pigs is not known. However, other reproductive performances such as gestation periods and litter sizes at birth and weaning in MMPs [18] were similar to those of other breeds of domestic pigs [15, 25, 26] including Göttingen and NIBS minipigs [27, 28]. Thus, apart from the farrowing rates, the MMP population used in the present study appears to have relatively normal porcine reproductive traits.

  1. Results, sections 3.3. and 3.4. The text of the two sections is rather difficult to read. To improve the readability, their introductory parts could be moved to Materials and Methods, section 2.4.; it is up to the authors to judge whether data in the tables 2 to 7 could be substituted with a text summary of the findings and whether the tables could be attached as supplementary materials. In my view, at least the addition of a clearly formulated summary of the data contained in the tables would significantly contribute to the readability of this part of the manuscript.

Response: According to the Reviewer’s suggestion, the introductory parts in sections 3.3. and 3.4. have been moved to Materials and Methods (lines 161-166). Furthermore, Tables 2, 3, 4, 6, and 7 have been attached as supplementary materials, Supplementary Tables 1, 2, 3, 4, and 5, respectively.

4   The Discussion as a whole is evidence-based and well written. I recommend adding information on possible interactions of the maternal NK cell receptor KIR and fetal HLA class I molecules in human reproduction in that part of Discussion dedicated to the biological context of the data presented (perhaps between lines 456-460).

Response: We appreciate your evaluation for our Discussion. According to the Reviewer’s suggestion, we have added two sentences on possible interactions of the maternal NK cell receptor KIR and fetal HLA class I molecules in Discussion section.

  1. Discussion (lines 528-535)

In humans, non-classical MHC class I molecules, HLA-E and HLA-G, can interact to maintain immune homeostasis of the maternal-fetus interface with many kinds of receptors on maternal immune cells in the uterus. HLA-E is the ligand of inhibitory and activating receptors, CD94/NKG2A and CD94/NKG2C, respectively, and HLA-G is the ligand of inhibitory receptors, ILT-2 (CD85j/LILRB1) and ILT-4 (CD85d/LILRB2) expressed on NK cells [41,42]. The pig appears to have many novel LILR genes, and only one KIR gene [43], but which of their gene products are expressed and might have a role in porcine reproduction is still not known.  

Reviewer 2 Report

In the manuscript “Genetic association between farrowing rates and swine leukocyte antigen alleles or haplotypes in Microminipigs”, Ando et al examined the potential associations between SLA haplotypes/alleles and farrowing rates in a highly inbred pig population called the Microminipigs. This is an interesting study with the aim of interrogating the impact of SLA diversity on specific reproductive performance. This study may aid our understanding of additional biological functions of SLA beyond their importance for the immune system.

Overall, the manuscript is fairly well written, concise and relatively easy to follow. The hypotheses, experiments and analyzes appeared to be scientifically and technically sound. However, the reviewer had some concerns and difficulties understanding the rationale behind how the data were analyzed and results were interpreted.

Given the fact that SLA genes are in a very tight linkage and that none of the 8 primary haplotypes share alleles (with the exception of DQB1 of 3 haplotypes), it was not clear on why the genes and alleles were analyzed separately as if they were independently assorted. For example, what were the rationales for not analyzing the complete SLA haplotype as a whole, or analyzing DRB1 separately from DQB1? Such granular analyses together with the relatively small sample size for each sharing/non-sharing combination would likely introduce biased results and skewed interpretations. The authors should provide the rationale behind and discuss the limitations and potential biases for such analytical approaches.

In addition, while it appeared that sharing/non-sharing of class I haplotypes had no impact on farrowing rates, it was not known for the class II haplotypes as the data was not analyzed in the same manner as the class I. The authors should clarify that and provide rationale for such independent analyses.

Furthermore, the frequency of the SLA haplotypes in the studied animal population were not equally distributed, with 3/8 SLA haplotypes accounting for more than 2/3 of the total haplotypes. In addition or as opposed to sharing/non-sharing of SLA, how would the authors be certain that the observed differences in farrowing rates were not contributed by individual SLA haplotypes, or other (non-SLA) genetic background of individual sows and/or boars? Such data and analyses were largely missing from the study.

The authors should clarify that this was a retrospective study of farrowing rates based on the animals’ SLA types. The authors should also provide more details on how the matings were accomplished (i.e. random vs selective) as it could have a major influence on the study outcomes.

There are a few spelling errors in the manuscript and would therefore benefit from additional spell check.

Author Response

For Reviewer 2

Comments and Suggestions for Authors

In the manuscript “Genetic association between farrowing rates and swine leukocyte antigen alleles or haplotypes in Microminipigs”, Ando et al examined the potential associations between SLA haplotypes/alleles and farrowing rates in a highly inbred pig population called the Microminipigs. This is an interesting study with the aim of interrogating the impact of SLA diversity on specific reproductive performance. This study may aid our understanding of

  1. Overall, the manuscript is fairly well written, concise and relatively easy to follow. The hypotheses, experiments and analyzes appeared to be scientifically and technically sound. However, the reviewer had some concerns and difficulties understanding the rationale behind how the data were analyzed and results were interpreted.

Response: To address the Reviewer’s comment, we have added following sentences in Materials and Methods.

  1. Materials and Methods

2.4. Influence of amino acid distance of SLA class I and class II genotypes between mating partners on farrowing rates (lines 161-166)

The sum of the amino acid pairwise distances among four alleles in each of the SLA class I genes and class II-DRB1 gene carrying by each mating pair were classified into nine to eleven groups that increased in distance from each other by 0.1 ranges. While on the other hand, the sum of the amino acid pairwise distances among the SLA four alleles of the class II- DQB1 gene carrying by each mating pair was classified into eight groups that increased in distance from each other by 0.05 ranges.

  1. Given the fact that SLA genes are in a very tight linkage and that none of the 8 primary haplotypes share alleles (with the exception of DQB1 of 3 haplotypes), it was not clear on why the genes and alleles were analyzed separately as if they were independently assorted. For example, what were the rationales for not analyzing the complete SLA haplotype as a whole, or analyzing DRB1 separately from DQB1? Such granular analyses together with the relatively small sample size for each sharing/non-sharing combination would likely introduce biased results and skewed interpretations. The authors should provide the rationale behind and discuss the limitations and potential biases for such analytical approaches.

 Response: The SLA genes are in a very tight linkage and that none of the 8 primary haplotypes share alleles except DQB1 alleles. Therefore, we have described the point in Results as follows,

  1. Results

3.1.  Association between farrowing rates and sharing of SLA class I haplotypes or alleles (lines 189-192)

Since associations among farrowing rates and sharing of SLA class I haplotypes amount to the same results as association among farrowing rates and sharing of SLA class I alleles, the following farrowing rates were represented mostly as sharing of SLA class I haplotypes.

We agree that analyses together with the relatively small sample size for each sharing/non-sharing combination would likely introduce biased results and skewed interpretations. We have added the possibilities of biased results and skewed interpretations in Discussion as follows,

  1. Discussion (lines 473-478)

 Nevertheless, we analyzed genetic association between farrowing rates and SLA alleles or haplotypes using relatively small number of MMPs in this study. Therefore, it might be difficult to exclude the possibilities of biased results in comparison of farrowing rates between sharing or non-sharing SLA alleles or haplotypes. The reliability of the present results could be confirmed in future studies by using a larger number of MMPs and other pig breeds.  

  1. In addition, while it appeared that sharing/non-sharing of class I haplotypes had no impact on farrowing rates, it was not known for the class II haplotypes as the data was not analyzed in the same manner as the class I. The authors should clarify that and provide rationale for such independent analyses.

 Response: As shown in Table 1, eleven SLA class I and II haplotypes including three recombinant haplotypes have been identified in the population of MMPs. Therefore, we have described the reason why we analyzed separately the average farrowing rates of mating partners with sharing SLA class I and class II haplotypes in Discussion as follows,

  1. Discussion (lines 460-464)

Due to the mating pairs with the recombinant haplotypes, the numbers of three mating groups on SLA class I haplotypes, CS, OS, and NS, were different from those on SLA class II haplotypes. Therefore, we analyzed separately the average farrowing rates of mating partners with sharing SLA class I and class II haplotypes.

  1. Furthermore, the frequency of the SLA haplotypes in the studied animal population were not equally distributed, with 3/8 SLA haplotypes accounting for more than 2/3 of the total haplotypes. In addition or as opposed to sharing/non-sharing of SLA, how would the authors be certain that the observed differences in farrowing rates were not contributed by individual SLA haplotypes, or other (non-SLA) genetic background of individual sows and/or boars? Such data and analyses were largely missing from the study.

 Response: As shown in Table 1, it is certainly true that eleven SLA haplotypes including three recombinant ones were not equally distributed, because the MMP population have not been bred by selectively matings based on SLA types with mating pairs. In fact, equable frequencies of SLA haplotypes in top three haplotypes, Hp-35.23, Hp-43.37, and Hp-17.17, or one of the three recombinant haplotypes, Hp-10.23 in dams, were not observed in the population. Regarding matings in the MMP population, we have also described in the below comment No. 5 of Reviewer 2.

Regarding analyses on association between farrowing rates and sharing/non-sharing of SLA using other breeds with different genetic background of individual sows and/or boars, unfortunately, we do not have any date using such other pig breeds or boar.

  1. The authors should clarify that this was a retrospective study of farrowing rates based on the animals’ SLA types. The authors should also provide more details on how the matings were accomplished (i.e. random vs selective) as it could have a major influence on the study outcomes.

 Response: Thank you for pointing that out. Regarding the matings of MMPs, we have added a following sentence in Materials and methods.

  1. Materials and methods

2.1 Animals (lines 91-94)

The matings of MMPs were basically random. However, during some generations especially with initial matings, mating pairs with relatively small body sizes were preferentially selected for matings to establish the characteristics of the MMP breed [19].

  1. There are a few spelling errors in the manuscript and would therefore benefit from additional spell check.

 Response: We have checked again spelling in the manuscript in detail and made some additional corrections.

Reviewer 3 Report

Dear authors,

This study analyses the effect of farrowing rates of SLA between mating partners in microminipigs. The relevant results indicate that SLA class I and II could use as genetic markers for selection in breeding programs in this specie. The manuscript is well written and structured, the introduction provides sufficient background and includes relevant references, the cited references are relevant to the research, the research design is appropriate, the methods are adequately described, and the conclusions are supported by the results. Only minor changes are necessary before publication. Concretely:

-          The figure 1 must be corrected, since it seems that it has been cut and the whole figure is not well seen.

-          In several parts of manuscript, the font is different (lines 267, 277, 279, and others).

-          The title of table 5 must be put in the journal format.

-          The position of tables and figures must be corrected. Each of them must be placed after being named in the text.

Author Response

For Reviewer 3フォームの始まり

Dear authors,

This study analyses the effect of farrowing rates of SLA between mating partners in microminipigs. The relevant results indicate that SLA class I and II could use as genetic markers for selection in breeding programs in this specie. The manuscript is well written and structured, the introduction provides sufficient background and includes relevant references, the cited references are relevant to the research, the research design is appropriate, the methods are adequately described, and the conclusions are supported by the results. Only minor changes are necessary before publication. Concretely:

-          The figure 1 must be corrected, since it seems that it has been cut and the whole figure is not well seen.

    Response: Due to the version of Microsoft word or Adobe pdf software, Figure 1 might be cut and the whole figure is not well seen. Since we have cited the figure 1 again, we would like to ask the editorial office whether it is well seen or not.   

-          In several parts of manuscript, the font is different (lines 267, 277, 279, and others).

     Response: Thank you for your advice. We have corrected the font on above positions and others.

-          The title of table 5 must be put in the journal format.

    Response: We have corrected the title of Table 2 in the journal format (original Table 5 has been corrected to Table 2 in our revised manuscript).

-          The position of tables and figures must be corrected. Each of them must be placed after being named in the text.

    Response: According to the Reviewer’s suggestion, we have precisely corrected the position of tables and figures. Table 1 and Figure 2 have been moved to lines 115 and 232, respectively.
